# Early Improvements in Disease Activity Indices Predict Long-Term Clinical Remission Suggested by the Treat-to-Target Strategy in Patients with Ankylosing Spondylitis Receiving TNF-α Inhibitor Treatment

**DOI:** 10.3390/jcm10184279

**Published:** 2021-09-21

**Authors:** Eon Jeong Nam, Won Kee Lee

**Affiliations:** 1Division of Rheumatology, Department of Internal Medicine, School of Medicine, Kyungpook National University, Daegu 41404, Korea; 2Medical Research Collaboration Center, School of Medicine, Kyungpook National University, Daegu 41944, Korea; wonlee@knu.ac.kr

**Keywords:** ankylosing spondylitis, TNF-α inhibitor, clinical remission, ASDAS-ID, BASDAI-CRP

## Abstract

This study evaluated the possibility of clinical remission suggested by the treat-to-target strategy and identified predictors of clinical remission in 139 patients with ankylosing spondylitis (AS) receiving tumor necrosis factor-α inhibitors (TNFi). Clinical remission criteria selected were AS Disease Activity Score Inactive Disease (ASDAS-ID) and Bath Ankylosing Spondylitis Disease Activity Index (BASDAI) < 2 with normal C-reactive protein (CRP) levels (BASDAI-CRP). The longitudinal relationship between clinical parameters and clinical remission was assessed using generalized estimating equations (GEEs). Responders to ASDAS-ID and BASDAI-CRP increased from 32.4% to 68.9% and from 39.9% to 75.2% at months 3 and 33, respectively. Responders to ASDAS-ID and BASDAI-CRP almost overlapped. In the univariable GEE model, age and 3-month improvement in BASDAI, ASDAS-CRP, physician and patient global assessments, and spinal pain predicted clinical remission achievement, while the presence of syndesmophytes predicted ASDAS-CRP achievement, and normalized CRP at 3 months was associated with BASDAI-CRP achievement. Multivariable GEE analysis revealed age (odds ratio (OR): 0.67; 95% confidence interval (CI), 0.49–0.93) and 3-month BASDAI improvement (OR: 1.70; CI, 1.19–2.41) as independent predictors of ASDAS-ID achievement and age (OR: 0.69; CI, 0.54–0.89), 3-month BASDAI improvement (OR: 2.00; CI, 1.45–2.76), and normalized CRP at 3 months (OR: 3.72; CI, 1.39–9.95) as independent predictors of BASDAI-CRP achievement.

## 1. Introduction

Over the past decade, tumor necrosis factor-α inhibitors (TNFi) have become the cornerstone for the treatment of ankylosing spondylitis (AS). These improve functional outcomes and decrease axial symptoms, peripheral arthritis, enthesitis, and extra-articular symptoms in patients with marginal benefit from non-steroidal anti-inflammatory drug (NSAID) therapy [1,2]. Furthermore, the Assessment of SpondyloArthritis International Society (ASAS)-European League Against Rheumatism (EULAR) [3] and American College of Rheumatology (ACR)/Spondylitis Association of America (SAA)/Spondyloarthritis Research and Treatment Network (SPARTAN) have recommended TNFi as the first biological agent in the treatment of AS [4,5].

As TNFi significantly improves clinical outcomes in patients with spondyloarthritis (SpA), treat-to-target (T2T) is an emerging management strategy for SpA [6,7,8]. The concept of T2T in SpA was developed based on the results of a systematic literature review and expert opinion by an international task force in 2012 and updated in 2017 [6,7]. According to this strategy, validated composite measures of disease activity, such as the AS Disease Activity Score (ASDAS) or Bath AS Disease Activity Index (BASDAI) with acute phase reactants (APRs), were recommended to be performed in order to measure clinical disease activity. The therapeutic target should be clinical remission, and low disease activity may be an alternative target. Clinical remission is mainly reflected by ASDAS-ID, low BASDAI with normal C-reactive protein (CRP) level, or ASAS partial remission (ASAS-PR).

For the measure of disease activity, ASDAS is a preferred method because it provides better disease discrimination than BASDAI and is more associated with various biomarkers of inflammation than BASDAI [6,7]. In case of BASDAI with APRs, BASDAI has some limitations in that its questions are only relevant to patient-reported outcomes and not specific for symptoms related to inflammation versus other processes [9]. Additionally, the definition of low BASDAI differs among studies due to the lack of a validated definition of clinical remission or low disease activity for BASDAI [10]. However, its ease of use, reproducibility, and sensitivity to changes make BASDAI a useful instrument for research purposes, and it has been used to assess the response to TNFi treatment in patients with AS [11]. Furthermore, APRs, which are objective measurements of inflammation, may complement BASDAI.

Many studies have evaluated the therapeutic effects of TNFi in patients with AS. Several studies that assessed predictors of clinical outcome in AS patients treated with TNFi used ASDAS-ID as the treatment target [12,13,14,15], and a few studies used low BASDAI with normal CRP as the target [13]. Although earlier studies identified only baseline clinical parameters as predictors for clinical remission [14,16,17], subsequent studies reported early clinical response as predictors of treatment outcome [12,13,15].

In this study, we aimed to evaluate the possibility of achieving clinical remission suggested by the T2T strategy, ASDAS-ID and low BASDAI with normal CRP, during long-term TNFi treatment and to investigate the predictors of clinical remission.

## 2. Materials and Methods

### 2.1. Patients

We performed a retrospective observational study on 139 patients with AS who were treated with TNFi for up to 33 months between January 2006 and December 2019. All patients were aged > 19 years, fulfilled the 1984 modified New York criteria for AS [18], and had begun TNFi therapy according to the Korean National Health Insurance guidelines, which recommend TNFi for use in patients who have active disease with BASDAI of ≥40/100, despite concurrent treatment with two or more anti-rheumatic drugs, including NSAIDs, for at least 3 months. At 3 months and every 6 months thereafter, TNFi treatment continuation was determined based on BASDAI improvement (50% improvement in BASDAI (BASDAI_50_) or absolute improvement of ≥2 units, compared with baseline), which was used as the definition of treatment response according to the Korean National Health Insurance guidelines. This study was approved by the Ethics Committee of the Kyungpook National University Hospital (KNUH 2020-12-030). Informed consent was waived due to the retrospective nature of the study.

### 2.2. Demographic and Clinical Data

Patients were evaluated at baseline, after 3 months, and then every 6 months for up to 33 months. Therapeutic targets that represented clinical remission in this study were ASDAS-ID and BASDAI < 2 with normal CRP level (BASDAI-CRP) criteria [6,13,19]. Disease activity was assessed using ASDAS based on CRP (ASDAS-CRP), BASDAI (0–10), physician global assessment (PhGA) (0–10), patient global assessment (PtGA) (0–10), spinal pain (0–10), and APRs (erythrocyte sedimentation rate (ESR) and CRP). ASDAS cut-offs for disease activity states are as follows: inactive disease (ASDAS-ID, ASDAS < 1.3), moderate disease activity (1.3 ≤ ASDAS < 2.1), high disease activity (2.1 ≤ ASDAS ≤ 3.5), and very high disease activity (ASDAS > 3.5) [20]. The ASDAS response criteria included ASDAS major improvement (ASDAS-MI, a change of ≥2.0 units, compared with baseline) and ASDAS clinically important improvement (ASDAS-CII, a change of ≥1.1 units, compared with baseline). BASDAI disease activity was defined using unvalidated cut-off values for low (BASDAI < 2), moderate (2 ≤ BASDAI < 4), high (4 ≤ BASDAI ≤ 6), or very high disease activity (BASDAI > 6), which are similar to those previously described in an observational study and a randomized controlled trial [13,19]. Physical functionality was assessed using the Bath AS Functional Index (BASFI). Spine and hip mobility were assessed using Bath Ankylosing Spondylitis Metrological Index (BASMI)_10_ and chest expansion.

### 2.3. Statistical Analysis

Descriptive statistics are expressed as mean ± standard deviation (SD). Continuous variables were compared using Student’s *t*-test and categorical variables using the chi-square or Fisher’s exact test. The paired *t*-test was used to compare outcome measures between visits. Bivariate correlations were determined using Spearman’s correlation coefficient. Longitudinal relationships between clinical parameters and achievement of clinical remission were assessed using generalized estimating equations (GEEs), with ASDAS-ID and BASDAI-CRP as dependent variables and individual clinical parameters as independent variables [21]. If a parameter demonstrated a significant association with the outcome in the univariable GEE model (*p* < 0.05), it was included as a covariate in the multivariable model. An “exchangeable” correlation structure was most appropriate for our study based on the correlation coefficients among outcomes at different follow-up evaluations [19]. Quasi-likelihood under the independence model criterion (QIC) was used to estimate the model fitness, and lower QICs reflected better data fit of the model. All statistical tests were performed at a significance level of α = 0.05. Statistical analyses were performed using SPSS (ver. 21.0; IBM, Armonk, NY, USA).

## 3. Results

### 3.1. Baseline Demographic and Clinical Characteristics

Baseline demographic and clinical characteristics of patients with AS are described in Table 1. A total of 139 patients treated with TNFi were included, with 54 (38.8%) receiving adalimumab, 54 (38.8%) receiving etanercept, 20 (14.4%) receiving golimumab, and 11 (7.9%) receiving infliximab. Seven patients who had discontinued previous TNFi due to inefficacy were included. During the study period, 34 patients (24.5%) discontinued TNFi treatment, 13 on adalimumab, 9 for etanercept, 11 for golimumab, and 1 for infliximab, with the reasons for discontinuation being adverse events (26.5%, *n* = 9), inefficacy (17.6%, *n* = 6), loss to followup due to change in residence (29.4%, *n* = 10), and patients’ intention (26.5%, *n* = 9).

Most patients were males (87.8%), and the mean age (SD) at the start of TNFi treatment was 37.5 (10.8) years. The mean disease duration was 9.9 (7.3) years, and syndesmophytes were detected in 53.2% of patients. The prevalence of human leukocyte antigen (HLA)-B27 was 93.5%, which was similar to that in other studies conducted in Eastern Asia [22,23]. Anterior uveitis was detected in 28.8% of patients. All patients demonstrated high or very high disease activity [19,20]. The mean ASDAS-CRP was 3.7 (0.8), with high disease activity (ASDAS > 2.1) in 94.2% and very high disease activity (ASDAS ≥ 3.5) in 58.3% patients; the mean BASDAI was 5.8 (1.2). The mean BASMI_10_ and chest expansion were 4.4 (2.2) and 2.8 (1.3), respectively.

### 3.2. Attainment Rate of Clinical Remission Criteria during TNFi Treatment

Percentages of ASDAS-ID responders were 32.4%, 46.8%, 50.8%, 55.8%, 55.4%, and 68.9% at 3, 9, 15, 21, 27, and 33 months, respectively, and the percentages of ASDAS-MI and ASDAS-CII responders ranged from 55.3% to 75.5% and from 88.6% to 96.9%, respectively. The attainment rate of BASDAI-CRP was 39.9%, 58.7%, 65.3%, 69.0%, 70.5%, and 75.2% at 3, 9, 15, 21, 27, and 33 months, respectively, and that of BASDAI_50_ ranged from 76.3% to 98.1% during the treatment period (Appendix A). The achievement rate of BASDAI-CRP was higher than that of ASDAS-ID over the TNFi treatment period, especially at 15 and 27 months (month 15, *p* = 0.035; month 27, *p* = 0.027) (Figure 1A). Patients who satisfied the ASDAS-ID and BASDAI-CRP almost overlapped. The proportion of patients who satisfied both clinical remission criteria also increased from 27.3% at 3 months to 63.8% at 33 months (Figure 1B). At 33 months, 94.4% of patients satisfying ASDAS-ID also satisfied BASDAI-CRP, and 84.8% of patients satisfying BASDAI-CRP satisfied ASDAS-ID (Figure 1B and Appendix A).

As ASDAS-ID and BASDAI-CRP do not have functional or metrological components, differences in the functional and metrological indices depending on satisfaction with clinical remission criteria were assessed. Responders to ASDAS-ID or BASDAI-CRP showed a significantly better functional condition than nonresponders when evaluated with BASFI (Appendix A). Furthermore, responders to ASDAS-ID or BASDAI-CRP demonstrated a significantly lower BASMI_10_ and higher chest expansion than nonresponders over the observational period. However, responders to BASDAI-CRP showed less prominent differences in metrological indices than compared to ASDAS-ID (Appendix A).

### 3.3. Early Improvement in Disease Activity Indices as a Predictor of ASDAS-ID Achievement during TNFi Treatment

We evaluated the predictors of ASDAS-ID achievement during TNFi treatment by using the GEE model (Table 2). The analysis using univariable GEE with baseline parameters, including age, disease duration, male sex, presence of HLA and syndesmophytes, and history of uveitis, demonstrated that older age (odds ratio (OR): 0.71, per 10 years increase; 95% confidence interval (CI), 0.54–0.92; *p* = 0.009) and presence of syndesmophytes (OR = 0.55; CI, 0.32–0.97; *p* = 0.038) were associated with a significantly lower possibility of achieving ASDAS-ID during TNFi treatment. Disease duration, presence of HLA-B27, and male sex did not influence the ASDAS-ID achievement.

As TNFi induced significant improvements in disease activity and functional and metrological indices with the most prominent change at 3 months (Appendix A), we conducted the univariable GEE models with 3-month improvements in these parameters to assess ASDAS-ID achievement. All 3-month improvements in disease activity indices, including ASDAS-CRP, BASDAI, PhGA, PtGA, and spinal pain, were associated with long-term ASDAS-ID achievement, whereas improvements in functional (BASFI) and metrological (BASMI_10_ and chest expansion) indices and APRs were not. Since age and the presence of syndesmophytes were predictors of ASDAS-ID achievement among baseline variables, 3-month improvements in these parameters were analyzed after adjustment for age and presence of syndesmophytes on the GEE model. All 3-month improvements in disease activity indices remained as significant variables: ASDAS-CRP (OR = 1.57; CI, 1.11–2.23; *p* = 0.011), BASDAI (OR = 1.78; CI, 1.40–2.26; *p* < 0.001), PhGA (OR = 1.47; CI, 1.26–1.71; *p* < 0.001), PtGA (OR = 1.40; CI, 1.26–1.71; *p* < 0.001), and spinal pain (OR = 1.32; CI, 1.13–1.53; *p* < 0.001). Normalized CRP, not ESR, at 3 months showed a marginally statistical significance (ESR, OR = 1.66; CI, 0.73–3.76; *p* = 0.226 and CRP, OR = 1.94; CI, 0.90–4.15; *p* = 0.089). The multivariable GEE analysis for ASDAS-ID achievement during TNFi treatment revealed age (OR = 0.67, per 10 years increase; CI, 0.49–0.93, *p* = 0.017) and 3-month BASDAI improvement (OR = 1.70; CI, 1.19–2.41; *p* = 0.003) to be associated with ASDAS-ID achievement, and the QIC was 816.657.

### 3.4. Early Change in Disease Activity Indices as a Predictor of BASDAI-CRP Attainment during TNFi Treatment

In the univariable GEE analysis using baseline variables, age (OR = 0.76, per 10 years increase; CI, 0.60–0.96; *p* = 0.021) was a negative predictor of BASDAI-CRP achievement (Table 3). The 3-month improvements in BASDAI (OR = 1.66; CI, 1.32–2.10; *p* < 0.001), PhGA (OR = 1.23; CI, 1.06–1.42; *p* = 0.006), PtGA (OR = 1.19; CI, 1.04–1.38; *p* = 0.015), spinal pain (OR = 1.25; CI, 1.09–1.45; *p* = 0.002), and normalized ESR (OR = 2.87; CI, 1.26–6.55; *p* = 0.012) or CRP (OR = 2.56; CI, 1.22–5.34; *p* = 0.048) at 3 months were identified as predictors of BASDAI-CRP achievement after adjustment for age. The 3-month ASDAS-CRP improvement was significantly associated with BASDAI-CRP achievement without adjustment for age (OR = 1.41; CI, 1.01–1.99; *p* = 0.047) and marginally associated with BASDAI-CRP achievement after adjustment (OR = 1.37; CI, 0.97–1.92; *p* = 0.074). The 3-month improvements in ESR, CRP, BASFI, and metrological indices did not predict BASDAI-CRP achievement.

Due to the fact that the proportion of normalized ESR or CRP at 3 months was similar (ESR, 85.8% (118/138); CRP, 87.0% (120/138)) and the 3-month improvement in ESR was significantly correlated with that in CRP (*r* = 0.712, *p* < 0.001), we performed multivariable GEE analysis for BASDAI-CRP achievement without the variable of normalized ESR at 3 months. On the multivariable GEE analysis, age (OR = 0.69, per 10 years increase; CI, 0.54–0.89); *p* = 0.004), 3-month BASDAI improvement (OR = 2.00; CI, 1.45–2.76; *p* < 0.001), and normalized CRP at 3 months (OR = 3.72; CI, 1.39–9.95; *p* = 0.009) were independent predictors of BASDAI-CRP achievement, and the QIC was 799.491.

### 3.5. Sub-Analysis of Identified Predictors for Achievement of Clinical Remission during TNFi Treatment

As younger age was associated with a higher achievement rate of both clinical remission criteria, we analyzed the proportion of attainment of ASDAS-ID and BASDAI-CRP using the age of 40 years as cut-off; patients aged < 40 years (*n* = 59) exhibited about two times the likelihood of achieving clinical remission than those aged ≥ 40 years (*n* = 80) (ASDAS-ID, OR = 2.05; CI, 1.16–3.62; *p* = 0.014 and BASDAI-CRP, OR = 1.99; CI, 1.17–3.37; *p* = 0.011). A significant difference in achievement rate based on the age of 40 years was observed at 9 months (*p* = 0.019), 15 (*p* = 0.005), and 27 (*p* = 0.037) in ASDAS-ID and at 3 months (*p* = 0.009) in BASDAI-CRP (Figure 2A,B).

In addition, as improvement in 3-month BASDAI scores showed association with achievement of treatment target at 33 months; subgroup analysis was performed after categorization according to the BASDAI scores. Patients with BASDAI improvement of ≥3 (*n* = 60) showed a higher achievement rate in both clinical remission criteria (ASDAS-ID, OR = 2.74; CI, 1.55–4.85; *p* = 0.001 and BASDAI-CRP, OR = 2.16; CI, 1.27–3.68; *p* = 0.004) than those with BASDAI improvement of <3 (*n* = 79), especially at 3 and 9 months (ASDAS-ID: 3 months, *p* < 0.001; 9 months, *p* < 0.001 and BASDAI-CRP: 3 months, *p* < 0.001; 9 months, *p* = 0.004) (Figure 2C,D). In the case of normalized ESR or CRP at 3 months, patients with normalized ESR or CRP at 3 months showed only a marginal significance in ASDAS-ID achievement (ESR, OR = 2.04; CI, 0.92–4.55; *p* = 0.080 and CRP, OR = 2.03; CI, 0.97–4.27; *p* = 0.060) (Figure 3A,C) but showed a significant difference in BASDAI-CRP attainment (ESR, OR = 3.29; CI, 1.42–7.46; *p* = 0.004 and CRP, OR = 2.73; CI, 1.32–5.62; *p* = 0.007), especially at 3 and 9 months (ESR: 3 months, *p* = 0.005; 9 months, *p* = 0.015 and CRP: 3 months, *p* < 0.001; 9 months, *p* = 0.021) (Figure 3B,D).

## 4. Discussion

In this study, the proportion of those who achieved BASDAI-CRP was higher than that of patients who achieved ASDAS-ID, and patients who satisfied the ASDAS-ID and BASDAI-CRP almost overlapped. In the multivariable GEE analysis, age and 3-month BASDAI improvement were independent predictors of the achievement of both clinical remission criteria. On the other hand, normalized CRP at 3 months was an independent predictor of BASDAI-CRP achievement. With respect to age, patients aged < 40 years showed two-times higher rate of achievement of both clinical remission criteria. For a 3-month BASDAI improvement ≥ 3, the probability of ASDAS-ID attainment increased by 2.7 times and that of BASDAI-CRP increased by 2.2 times.

The concept of T2T strategies has been widely used to obtain optimal outcomes in chronic non-rheumatic diseases, such as hypertension and hyperlipidemia [24]. T2T strategies have also been incorporated in the management of rheumatic diseases, especially rheumatoid arthritis (RA) [8,25]. This treatment approach in RA is now a well-established practice with regular disease activity monitoring using validated composite measures of disease activity and remission criteria. The T2T approach in SpA was recommended by an international task force [6], but it has not yet been established because of the lack of robust direct evidence and no absolute consensus about which therapeutic target to use [5,8].

Although there is no widely accepted definition for clinical remission in the T2T strategy in AS, ASDAS-ID was suggested by the task force as it comprises the estimation of clinical aspects of the disease as well as APRs [6,7,8]. Low BASDAI with normal CRP level and ASAS-PR were alternatively suggested as clinical remission criteria [6,7]. We selected both ASDAI-CRP and BASDAI-CRP as clinical remission criteria because BASDAI incorporates additional clinical outcomes of interest, such as physical function; assesses enthesopathy; and has both eligibility criterion and outcome in clinical trials [9]. In addition, in clinical practice, the BASDAI measurement is an easier striker compared to the ASAS-PR assessment. Moreover, CRP, an objective parameter, may compensate for some of the shortcomings of BASDAI. Due to the unvalidated definition of low disease activity or clinical remission for BASDAI [10], different cut-off values have been used in observational [26], cohort [19], and controlled [13,27] studies as follows: BASDAI < 4 [26], BASDAI < 3 [27], and BASDAI < 2 [13,19]. In this study, we adopted the strictest cut-off value, BASDAI < 2, which was used in cohort [19] and randomized controlled [13] studies. We found patients who satisfied the ASDAS-ID, and BASDAI-CRP almost overlapped, but the achievement rate of BASDAI-CRP was higher than that of ASDAS-ID. This may be because both clinical remission criteria share common components, such as CRP and variables of back pain, duration of morning stiffness, and peripheral joint symptoms in BASDAI, whereas ASDAS-ID achievement needs PtGA improvement as another component. Furthermore, BASDAI < 2 as the cut-off value may not imply an inactive disease status [11].

Early studies demonstrated that younger age, male sex, HLA-B27 positivity, higher disease activity, increased CRP, and higher functional status were independent baseline predictors of the clinical response to TNFi treatment [14,16,17]. Subsequent studies showed that early response to TNFi predicted long-term clinical improvement [12,13,15], and the effect of baseline characteristics on the long-term clinical improvement was relatively small compared with the effect of early clinical response [12].

In our study, among baseline demographic factors, age was an independent predictor of achievement of both clinical remission criteria. Older patients usually had longer disease recovery periods, while younger patients tended to have shorter disease duration. Age was closely correlated with disease duration (*r* = 0.569, *p* < 0.001). These results suggest that because young age is related with early diagnosis, young patients have a great effect on TNFi treatment. However, disease duration did not predict clinical remission achievement in our study. Several studies also demonstrated that age was associated with TNFi response, whereas disease duration was not [14,16,28], which suggests that there may be other mechanisms in the relationship between age and TNFi response. One suggested underlying explanation is that young age is related with low percentage of TNFα-producing CD8+ T cells in AS [28]. Absence of syndesmophytes was also a positive predictor of ASDAS-ID achievement. Disease duration was not associated with the achievement of clinical remission criteria, which may be due to the long disease duration of the enrolled patients in our study. As the clinical improvements at 3 months were the most prominent with TNFi treatment in our study, we tried to demonstrate that early clinical improvements were associated with a higher possibility of sustained attainment of long-term clinical remission. In terms of achievements of both clinical remission criteria, the 3-month BASDAI improvement was the strongest predictor. Higher 3-month BASDAI improvement, especially BASDAI improvement of ≥3, was an independent predictor of achievement of long-term clinical remission criteria in both ASDAS-ID and BASDAI-CRP. On the other hand, the 3-month ASDAS-CRP improvement was not a strong predictor of achievement of clinical remission criteria on the univariable GEE model and was not an independent predictor on multiple GEE model. Considering that ASDAS-CRP shares components of BASDAI, multivariable GEE model without the 3-month BASDAI improvement for ASDAS-ID achievement was also performed. However, even in this case, the 3-month PhGA, not the 3-month ASDAS-CRP improvement, was an independent predictor of ASDAS-ID achievement (data not shown). Normalized CRP at 3 months was an independent predictor of BASDAI-CRP, not ASDAS-ID, achievement, which may be related to the fact that BASDAI-CRP achievement needs both CRP normalization and low BASDAI, while ASDAS-ID achievement demands not only CRP improvement but also improvements in other four variables. With respect to the 3-month improvements, PhGA, PtGA, and spinal pain were associated with the attainment of ASDAS-ID and BASDAI-CRP in the univariable GEE model but not in the multivariable GEE model. It is possible that the interaction of these three factors resulted in a loss of statistical significance. The 3-month improvements in functional (BASFI) and metrological (BASMI_10_ and chest expansion) indices were not associated with long-term clinical remission achievement, which may be related with irreversible structural changes due to longer disease duration.

Our study has some limitations. First, because we selected BASDAS < 2 as the cut-off for clinical remission for BASDAI, analysis using different cut-offs may result in divergence of results in identifying the predictors of attainment of clinical remission. Second, because this was a retrospective and observational study, bias may be present. For example, if a description of the reason for elevated CRP was lacking, whether it was caused by disease activity or by accompanying infection, it had to be inferred by using other disease activity indices. Further studies are required to determine whether BASDAI-CRP achievement can be a clinical remission criterion in T2T strategy after the validation of the cut-off value of low disease activity in BASDAI during long-term TNFi treatment in AS.

In conclusion, we found that the proportion of patients who achieved BASDAI-CRP was higher than those who achieved ASDAS-ID, and patients who satisfied the ASDAS-ID and BASDAI-CRP almost overlapped. Age and 3-month BASDAI improvement were independent predictors of achievement of both long-term clinical remission criteria, and the normalized CRP at 3 months was an independent predictor of BASDAI-CRP achievement.

## Figures and Tables

**Figure 1 jcm-10-04279-f001:**
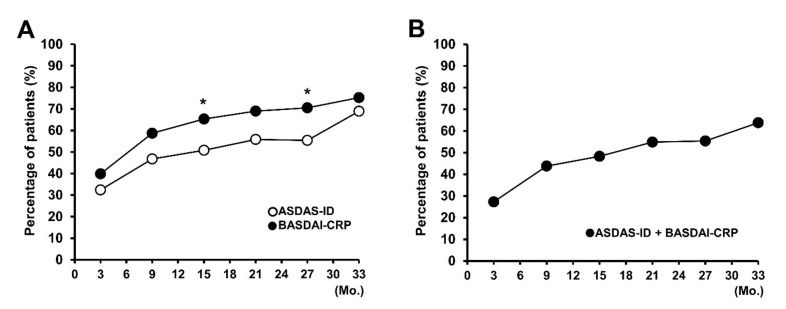
Response to TNF-α-inhibitor (TNFi) treatment as assessed by the ASDAS inactive disease (ASDAS-ID) and low BASDAI with normal CRP level (BASDAI-CRP). These graphs show the proportion of patients who achieved (**A**) ASDAS-ID or BASDAI-CRP and (**B**) both ASDAS-ID and BASDAI-CRP. * *p* < 0.05. TNF, tumor necrosis factor; ASDAS, Ankylosing Spondylitis Disease Activity Score; BASDAI, Bath Ankylosing Spondylitis Disease Activity Index; CRP, C-reactive protein.

**Figure 2 jcm-10-04279-f002:**
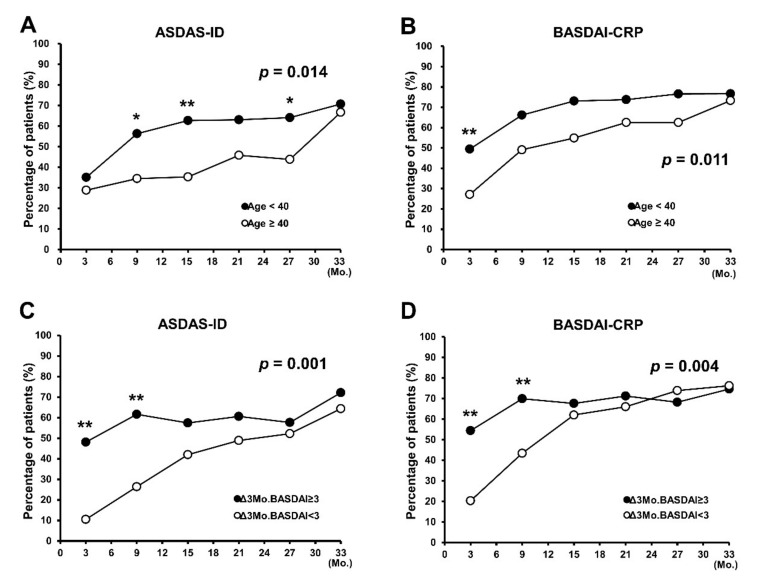
Sub-analysis of predictors of clinical remission achievement during TNF-α inhibitor treatment. (**A**,**B**) Analysis of the attainment rate of (**A**) ASDAS-ID and (**B**) BASDAI-CRP according to the age of 40 years. (**C**,**D**) Analysis of the achievement rate of (**C**) ASDAS-ID and (**D**) BASDAI-CRP according to BASDAI of 3. ASDAS-ID, Ankylosing Spondylitis Disease Activity Score Inactive Disease; BASDAI-CRP, low Bath Ankylosing Spondylitis Disease Activity Index with Normal C-reactive protein level; Δ3Mo.BASDAI, change in BASDAI from baseline to 3 months; TNF, tumor necrosis factor. * *p* < 0.05, ** *p* < 0.01.

**Figure 3 jcm-10-04279-f003:**
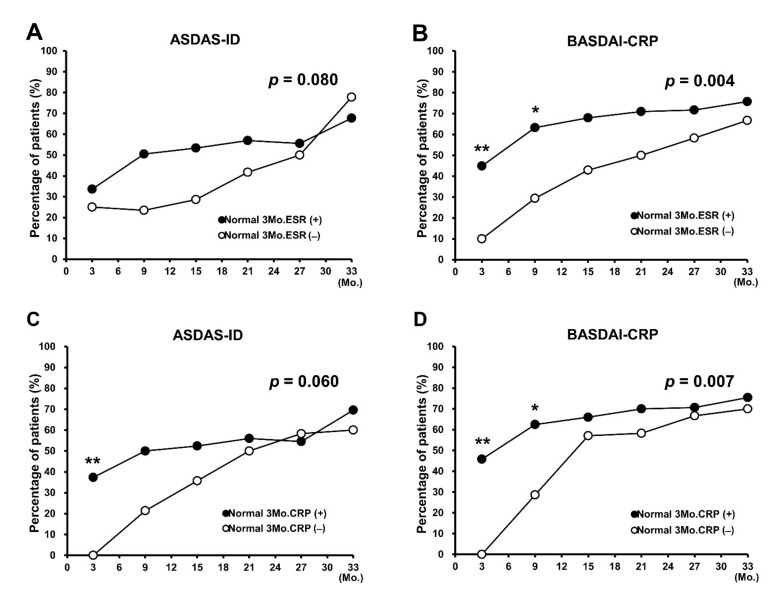
Subanalysis of predictors of clinical remission achievement during TNF-α inhibitor treatment. (**A**,**B**) Analysis of the attainment rate of (**A**) ASDAS-ID and (**B**) BASDAI-CRP according to normalized ESR at 3 months. (**C**,**D**) Analysis of the achievement rate of (**C**) ASDAS-ID and (**D**) BASDAI-CRP according to normalized CRP at 3 months. ASDAS-ID, Ankylosing Spondylitis Disease Activity Score Inactive Disease; BASDAI-CRP, low Bath Ankylosing Spondylitis Disease Activity Index with normal C-reactive protein level; normal 3Mo.ESR, normalized ESR at 3 months; normal 3Mo.CRP, normalized CRP at 3 months; TNF, tumor necrosis factor. * *p* < 0.05, ** *p* < 0.01.

**Table 1 jcm-10-04279-t001:** Baseline demographic and clinical characteristics of patients with ankylosing spondylitis.

Characteristics	Results
Patient (male, %)	139 (122, 87.8)
Age, years	37.5 ± 10.8
Disease duration	9.9 ± 7.3
HLA-B27 positive	116/124 (93.5)
Uveitis	40 (28.8)
ASDAS-CRP	3.7 ± 0.8
BASDAI (0–10)	5.8 ± 1.2
PhGA (0–10)	6.3 ± 1.6
PtGA (0–10)	6.6 ± 1.7
Spinal pain (0–10)	6.5 ± 1.7
BASFI (0–10)	4.4 ± 2.2
BASMI_10_	4.1 ± 2.3
Chest expansion	2.8 ± 1.3
ESR (mm/h)	36.3 ± 25.1
CRP (mg/dL)	2.5 ± 2.3
Syndesmophytes (≥1)	74 (53.2)
Used drugs	
Adalimumab	54 (38.8)
Etanercept	54 (38.8)
Golimumab	20 (14.4)
Infliximab	11 (7.9)

Values are presented as the number of patients (%) or mean ± standard deviation (SD). HLA, human leucocyte antigen; ASDAS-CRP, Ankylosing Spondylitis Disease Activity Score based on CRP; BASDAI, Bath Ankylosing Spondylitis Disease Activity Index; PhGA, physician global assessment; PtGA, patient global assessment; BASFI, Bath Ankylosing Spondylitis Functional Index; BASMI, Bath Ankylosing Spondylitis Metrological Index.

**Table 2 jcm-10-04279-t002:** Effects of improvement in disease activity indices at 3 months on the achievement of Ankylosing Spondylitis Disease Activity Score Inactive Disease (ASDAS-ID) during the treatment with tumor necrosis factor-α inhibitor in patients with ankylosing spondylitis.

Variable	Univariable Model, OR (95% CI)	Univariable Model Adjusted for Age and Syndemophytes *	Multivariable Model, OR (95% CI)
Age, 10 years	**0.71 (0.54–0.92)**	-	**0.67 (0.49–0.93**)
Disease duration, years	0.97 (0.93–1.00)	-	†
Sex, male	0.67 (0.28–1.60)	-	†
HLA-B27 (+)	0.68 (0.25–1.84)	-	†
Syndesmophytes (+)	**0.55 (0.32–0.97)**	-	0.75 (0.36–1.58)
Uveitis (+)	0.83 (0.45–1.54)	-	†
Δ3Mo.ASDAS-CRP	**1.68 (1.17–2.40)**	**1.57 (1.11–2.23)**	0.66 (0.37–1.15)
Δ3Mo.BASDAI	**1.76 (1.36–2.27)**	**1.78 (1.40–2.26)**	**1.70 (1.19–2.41)**
Δ3Mo.PhGA	**1.47 (1.26–1.72)**	**1.47 (1.26–1.71)**	1.28 (0.95–1.74)
Δ3Mo.PtGA	**1.38 (1.19–1.61)**	**1.40 (1.21–1.63)**	1.16 (0.85–1.60)
Δ3Mo.Pain	**1.31 (1.12–1.52)**	**1.32 (1.13–1.53)**	0.89 (0.67–1.18)
Δ3Mo.BASFI	1.13 (0.95–1.34)	1.15 (0.97–1.35)	†
Δ3Mo.BASMI_10_	1.31 (0.95–1.80)	1.27 (0.92–1.75)	†
Δ3Mo.ChE	1.15 (0.88–1.51)	1.09 (0.83–1.42)	†
Δ3Mo.ESR	1.01 (0.99–1.02)	1.00 (0.99–1.01)	†
Δ3Mo.CRP	1.06 (0.93–1.20)	1.05 (0.94–1.18)	†
Normal 3Mo.ESR	2.04 (0.92–4.55)	1.66 (0.73–3.76)	†
Normal 3Mo.CRP	2.03 (0.97–4.27)	1.94 (0.90–4.15)	†
QIC of the model			816.657

Odds ratios (ORs) for potential predictors of ASDAS-ID achievement were analyzed using a generalized estimating equation model. Bold type indicates *p* < 0.05. * The model was adjusted for baseline parameters showing relevant association (*p* < 0.05) with the outcome on the univariable analysis. † Not included in the model because its association with the outcome was not relevant (*p* ≥ 0.05). ASDAS-ID, Ankylosing Spondylitis Disease Activity Score Inactive Disease; OR, odds ratio; CI, confidence interval; HLA-B27, human leucocyte antigen B27; Δ3Mo.ASDAS-CRP, change in Ankylosing Spondylitis Disease Activity Score based on CRP from baseline to 3 months; Δ3Mo.BASDAI, change in Bath Ankylosing Spondylitis Disease Activity Index from baseline to 3 months; Δ3Mo.PhGA, change in physician global assessment from baseline to 3 months; Δ3Mo.PtGA, change in patient global assessment from baseline to 3 months; Δ3Mo.Pain, spinal pain change from baseline to 3 months; Δ3Mo.BASFI, change in BASFI from baseline to 3 months; Δ3Mo.BASMI_10_, change in BASMI_10_ from baseline to 3 months; Δ3Mo.ChE, change in chest expansion from baseline to 3 months; Δ3Mo.ESR, change in ESR from baseline to 3 months; Δ3Mo.CRP, change in CRP from baseline to 3 months; normal 3Mo.ESR, normalized ESR at 3 months; normal 3Mo.CRP, normalized CRP at 3 months; QIC, quasi-likelihood under the independent model criterion.

**Table 3 jcm-10-04279-t003:** Effect of improvement in disease activity indices at 3 months on the achievement of low Bath Ankylosing Spondylitis Disease Activity Index and normal C-reactive protein level (BASDAI-CRP) during the treatment with tumor necrosis factor-α inhibitor in patients with ankylosing spondylitis.

Variable	Univariable Model, OR (95% CI)	Univariable Model Adjusted for Age *	Multivariable Model, OR (95% CI)
Age, 10 years	**0.76 (0.60–0.96)**	-	**0.69 (0.54–0.89)**
Disease duration, years	0.99 (0.95–1.02)	-	†
Sex, male	0.85 (0.41–1.75)	-	†
HLA-B27 (+)	0.99 (0.33–2.93)	-	†
Syndesmophytes (+)	0.82 (0.48–1.39)	-	†
Uveitis (+)	0.90 (0.50–1.62)	-	†
Δ3Mo.ASDAS-CRP	1.41 (1.01–1.99)	1.37 (0.97–1.92)	0.56 (0.30–1.03)
Δ3Mo.BASDAI	1.64 (1.30–2.06)	**1.66 (1.32–2.10)**	**2.00 (1.45–2.76)**
Δ3Mo.PhGA	1.23 (1.06–1.43)	**1.23 (1.06–1.42)**	1.04 (0.79–1.37)
Δ3Mo.PtGA	1.18 (1.01–1.37)	**1.19 (1.04–1.38)**	0.91 (0.68–1.23)
Δ3Mo.Pain	1.24 (1.08–1.43)	**1.25 (1.09–1.45)**	1.13 (0.86–1.48)
Δ3Mo.BASFI	1.03 (0.89–1.20)	1.05 (0.91–1.22)	†
Δ3Mo.BASMI_10_	1.25 (0.91–1.71)	1.25 (0.92–1.69)	†
Δ3Mo.ChE	1.11 (0.84–1.46)	1.06 (0.80–1.41)	†
Δ3Mo.ESR	1.01 (1.00–1.02)	1.00 (0.99–1.02)	†
Δ3Mo.CRP	1.06 (0.95–1.19)	1.05 (0.94–1.16)	†
Normal 3Mo.ESR	3.29 (1.45–7.46)	**2.87 (1.26–6.55)**	‡
Normal 3Mo.CRP	2.73 (1.32–5.62)	**2.56 (1.22–5.34)**	**3.72 (1.39–9.95)**
QIC of the model			799.491

ORs for potential predictors of BASDAI-CRP achievement were analyzed using a generalized estimating equation model. Bold type indicates *p* < 0.05. * The model was adjusted for baseline parameters showing relevant association (*p* < 0.05) with the outcome on the univariable analysis. † Not included in the model because its association with the outcome was not relevant (*p* ≥ 0.05). ‡ Not included in the model because the proportion of normalized ESR or CRP at month 3 was similar (ESR, 85.8% (118/138); CRP, 87.0% (120/138)), and 3-month improvement in ESR was significantly correlated with that in CRP (*r* = 0.712, *p* < 0.001). BASDAI-CRP, BASDAI < 2 with normal CRP; OR, odds ratio; CI, confidence interval; HLA-B27, human leucocyte antigen B27; Δ3Mo.ASDAS-CRP, change in Ankylosing Spondylitis Disease Activity Score based on CRP from baseline to 3 months; Δ3Mo.BASDAI, change in Bath Ankylosing Spondylitis Disease Activity Index from baseline to 3 months; Δ3Mo.PhGA, change in physician global assessment from baseline to 3 months; Δ3Mo.PtGA, change in patient global assessment from baseline to 3 months; Δ3Mo. Pain, spinal pain change from baseline to 3 months; Δ3Mo.BASFI, change in BASFI from baseline to 3 months; Δ3Mo.BASMI_10_, change in BASMI_10_ from baseline to 3 months; Δ3Mo.ChE, change in chest expansion from baseline to 3 months; Δ3Mo.ESR, change in ESR from baseline to 3 months; Δ3Mo.CRP, change in CRP from baseline to 3 months; normal 3Mo.ESR, normalized ESR at 3 months; normal 3Mo.CRP, normalized CRP at 3 months; QIC, quasi-likelihood under the independent model criterion.

## Data Availability

All data generated or analyzed during this study are included in the study.

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
