# Peer review of "Early Improvements in Disease Activity Indices Predict Long-Term Clinical Remission Suggested by the Treat-to-Target Strategy in Patients with Ankylosing Spondylitis Receiving TNF-α Inhibitor Treatment"

_jcm, 2021, doi:10.3390/jcm10184279_

Round 1
Reviewer 1 Report
It's a well-written article. Please address the following comments
Can the authors explain the improvement in the young age and is it related to early diagnosis?
Please explain why you choose anti - TNF in this study? why the authors did not compare anti TNF with other biologicals?
It is been well established anti - TNF is the first-line medication for spondyloarthropathy? What additional information this does the article provide?
Author Response
Response to the Reviewer 1
We thank you and the reviewer for the constructive comments on our manuscript. We have addressed all of the points made by the reviewers and provided specific rebuttals in a point-by-point manner. Furthermore, supplementary information has been presented in the responses to reviewer comments. We hope that the revised manuscript is now suitable for publication in Journal of Clinical Medicine.
Reviewer 1.
- Can the authors explain the improvement in the young age and is it related to early diagnosis?
[Response] We appreciate the reviewer’s suggestion. In this study, the mean age at the start of TNF-α inhibitor (TNFi) was 37.5 years (median [quartile]: 37.0, 30-45 years; the youngest patient was 19 years old and the oldest was 69). The mean disease duration was 9.9 years (median [quartile]: 9.0, 4.25-14 years; the lowest was 0.5 years and the highest was 40 years). Older patients usually had longer disease recovery period, while younger patients tended to have shorter disease duration. Age was closely correlated with disease duration (r = 0.569, p <0.001). These results suggest that because young age is related with early diagnosis, young patients have a great effect on TNFi treatment. However, disease duration did not predict clinical remission achievement in our study. Several studies also demonstrated that age was associated with TNFi response, whereas disease duration was not [1,2,3], which suggests that there may be other mechanisms in the relationship between age and TNFi response. One suggested underlying explanation is that young age is related with low percentage of TNFα-producing CD8+ T cells in ankylosing spondylitis (AS) [3].
We have updated our manuscript with additional data in the “Discussion” section as follow: “In our study, among baseline demographic factors, age was an independent predictor of achievement of both clinical remission criteria. Older patients usually had longer disease recovery period, while younger patients tended to have shorter disease duration. Age was closely correlated with disease duration (r = 0.569, p <0.001). These results suggest that because young age is related with early diagnosis, young patients have a great effect on TNFi treatment. However, disease duration did not predict clinical remission achievement in our study. Several studies also demonstrated that age was associated with TNFi response, whereas disease duration was not, which suggests that there may be other mechanisms in the relationship between age and TNFi response. One suggested underlying explanation is that young age is related with low percentage of TNFα-producing CD8+ T cells in AS.”
- Please explain why you choose anti - TNF in this study? why the authors did not compare anti TNF with other biologicals?
[Response] We appreciate the reviewer’s comment. There have been major advances in the management of axial spondyloarthritis (axSpA) with the introduction of effective biologic agents targeting TNF-α and IL-17A. The current licensed drug treatments for axial spondyloarthritis (axSpA) are NSAIDs and biologic DMARDs targeting TNF-α or IL-17A [4]. TNFi, including etanercept, infliximab, adalimumab, certolizumab pegol, and golimumab, are biologic agents which are FDA-approved to treat AS. Etanercept received FDA approval for treatment of AS in 2003 and was first FDA-approved biologic for treatment of AS. Since then, other TNFi also have been used to treat AS with FDA approval. TNFi have become the cornerstone for the treatment of AS. The first IL-17A inhibitor, secukinumab, obtained FDA approval as treatment of AS in 2016 and has been used as second-line biologic in the event of TNFi failure in Korea. Therefore, the number of patients receiving IL-17A inhibitor is small and we could not conduct a study on the efficacy comparison between TNFi with IL-17 inhibitor in AS.
- It has been well established anti-TNF is the first-line medication for spondyloarthropathy? What additional information this does the article provide?
[Response] Thank you for your comments. Biologic therapies, including TNFi and IL-17A inhibitors, have vastly improved clinical outcomes for patients with spondyloarthritis (SpA). Consequently, targeting clinical remission/inactive disease is now a major treatment goal as outlined in current treat-to-target (T2T) recommendations [5]. The treat-to-target (T2T) is emerging management strategy in SpA. Although evidence from randomised controlled trials for a T2T approach in AS is still lacking, observational evidence suggests that a T2T approach might be beneficial in AS [6]. Therapeutic targets in T2T strategy have been defined for AS, based on the results of a systematic literature review and expert opinion, and should be clinical remission/inactive disease of musculoskeletal and extra-articular manifestation [7,8].
In this study, we evaluated the possibility of achieving clinical remission suggested by the T2T strategy, ASDAS inactive disease (ASDAS-ID) and low BASDAI with normal CRP (BASDAI-CRP), during long-term TNFi treatment and to investigate the predictors of clinical remission. We found that the proportion of patients who achieved BASDAI-CRP was higher than that of those who achieved ASDAS-ID and age and 3-month BASDAI improvement were independent predictors of both long-term clinical remission achievement, and normalized CRP at 3 months was an independent predictor of BASDAI-CRP achievement. Our results suggest that ASDAS-ID is more stringent than BASDAI-CRP and age and early clinical improvements, especially BASDAI improvement, are associated with clinical remission achievement.
References
- Vastesaeger, N.; van der Heijde, D.; Inman, R.D.; et al. Predicting the outcome of ankylosing spondylitis therapy. Ann Rheum Dis 2011, 70, 973-981.
- Arends, S.; Brouwer, E.; van der Veer, E.; et al. Baseline predictors of response and discontinuation of tumor necrosis factor-alpha blocking therapy in ankylosing spondylitis: a prospective longitudinal observational cohort study. Arthritis Res Ther 2011,13(3):R94.
- Schramm-Luc, A.; Schramm, J.; Siedliński, M.; Guzik, T.J.; Batko, B. Age determines response to anti-TNFα treatment in patients with ankylosing spondylitis and is related to TNFα-producing CD8 cells. Clin Rheumatol 2018, 37(6),1597-1604.
- Fragoulis GE, Siebert S. Treatment strategies in axial spondyloarthritis: what, when and how? Rheumatology (Oxford) 2020, 59(Suppl4), iv79-iv89.
- Baraliakos, X.; Berenbaum, F.; Favalli, E.G.; et alChallenges and Advances in Targeting Remission in Axial Spondyloarthritis. J Rheumatol 2018, 45(2),153-157.
- Smolen, J.S. Treat-to-target as an approach in inflammatory arthritis. Curr Opin Rheumatol 2016, 28, 297-302.
- Smolen, J.S.; Braun, J.; Dougados, M.; et al. Treating spondyloarthritis, including ankylosing spondylitis and psoriatic ar-thritis, to target: recommendations of an international task force. Ann Rheum Dis 2014, 73, 6-16.
Smolen, J.S.; Schols, M.; Braun, J.; et al. Treating axial spondyloarthritis and peripheral spondyloarthritis, especially psori-atic arthritis, to target: 2017 update of recommendations by an international task force. Ann Rheum Dis 2018, 77, 3-17.
Reviewer 2 Report
Great effort and great research question.
Please consider improving discussion section. Other changes are suggested in the file attached.

Author Response
Response to the Reviewer 2
We thank you and the reviewer for the constructive comments on our manuscript. Although English language editing was done before submitting this manuscript, we are sorry for the fact that there are many amendments. We have addressed all of the points made by the reviewers and provided specific rebuttals in a point-by-point manner. Furthermore, supplementary information has been presented in the responses to reviewer comments. We hope that the revised manuscript is now suitable for publication in Journal of Clinical Medicine.
- Commented [A1]: ASDAS includes CRP as a biomarker. According to the reference cited here, ASDAS is more associated with various biomarkers of inflammation than BASDAI. That is correct. However, authors wrote BASDAI plus CRP in the sentence which makes it confusing and wrong. I agree with first part of the sentence where you said ASDAS is more discriminative than BASDAI plus CRP.
[Response] We appreciate the reviewer’s critical comments. We have revised the sentence as per the reviewer’s suggestion as follows; “For measure of disease activity, ASDAS is a preferred method because it provides better disease discrimination than BASDAI and is more associated with various biomarkers of inflammation than BASDAI.”
- Commented [A2]: Consider changing the term "recommended" to "used"
[Response] Thank you for the suggestion. We have revised the sentence as per the reviewer’s suggestion as follows; “However, its ease of use, reproducibility, and sensitivity to changes make BASDAI a useful instrument for research purposes, and it has been used to assess the response to TNFi treatment in patients with AS.”
- Commented [A3]: Several studies that assessed predictors of clinical outcome in AS patients treated with TNFi used ASDAS-ID as the treatment target and a few used low BASDAI with normal CRP as the target.
[Response] Thank you for the suggestion. We have revised the sentence as per the reviewer’s suggestion as follows: “Several studies that assessed predictors of clinical outcome in AS patients treated with TNFi used ASDAS-ID as the treatment target, and a few used low BASDAI with normal CRP as the target.”
- Commented [A4]: Please correct this sentence to make the idea clear. One suggestion is to say-Although earlier studies identified only baseline clinical parameters as predictors for clinical remission, subsequent studies reported early clinical response as predictors of treatment outcome.
[Response] We appreciate the reviewer’s comment. We have revised the sentence as per the reviewer’s suggestion as follows; “Although earlier studies identified only baseline clinical parameters as predictors for clinical remission, subsequent studies reported early clinical response as predictors of treatment outcome.”
- Commented [A5]: Was an IRB approval obtained?
[Response] Thank you for the comments. It is a retrospective study that analyzes the medical records while the medical treatment has already been completed in December 2019. Therefore, it is practically impossible to obtain consent from the subjects. The risk of personal information leakage is very low because the persons in charge of the research are kept accessible by encrypting the patient’s personal identification information. In addition, there is no direct gain or loss to the enrolled subjects depending on their participation. Based on the above, the authors submitted a consent exemption documents and obtained approval from the IRB (KNUH 2020-12-030).
- Commented [A11]: There is an overrepresentation of HLA B27 positive patients. it is worth mentioning that under Discussion. / Commented [A13]: There is an overrepresentation of HLA B27 positive patients in this cohort.
[Response] Thank you for the valuable comment. The association between HLA-B27 and the AS is so strong that the HLA-B27 molecule has been speculated to play a direct pathogenetic role [1]. The prevalence of HLA-B27 is reported 60 to 95% [2-5]. In the studies of European patients with AS, the prevalence of HLA-B27 in patients with advanced AS is estimated to be ~80% in the German Spondyloarthritis Inception Cohort [2] and is 80 to 95 percent of white (western European descent) patients with AS [3]. In the studies of Asian patients with AS, the prevalence of HLA-B27 in Chinese patients with AS is about 95% [4]. In Korea, Kim et al. reported that 94.8% of patients were HLA-B27 carrier in 840 Korean patients with AS [5]. The prevalence of HLA-B27 in our study was 93.7%, which was similar to that in other studies conducted in eastern Asia.
We have updated our manuscript with additional description in the “Results” section as follows; “The prevalence of human leukocyte antigen (HLA)-B27 was 93.5%, which was similar to that in other studies conducted in eastern Asia.”
- Commented [A12]: are they redundant (? not useful) / Commented [A18]: ?redundant / Commented [A37]: ?redundant Commented [A45]: ?redundant
[Response] Thank you for the comment. We have revised the sentence as follows; “Patients who satisfied the ASDAS-ID and BASDAI-CRP almost overlapped.” and “Responders to ASDAS-ID and BASDAI-CRP almost overlapped.”
- Commented [A14]: Could this be due to longer disease duration (9years) that resulted in irreversible structural changes?
[Response] Thank you for your comments. AS is a linearly progressive disease with about 35% change every 10 years. Spinal involvement is largely an expression of disease duration [6]. Disease activity contributes longitudinally to radiographic progression in the spine in AS. This effect is more pronounced in the earlier phases of the disease [7]. Mobility in AS, which is measured by Bath AS Metrological Index (BASMI), is influenced by both structural damage and activity, but definitely also by age and disease duration [8]. The BASMI correlates moderately with changes in functional outcomes as measured by the Bath AS Functional Index (BASFI) (r = 0.46 for BASMI10, P <0.001) [9]. In our study, the mean disease duration was 9.9 years and one or more syndesmophytes were detected in 53.2% of patients. The 3-month improvements in functional index (BASFI) and metrological indices (BASMI10 and chest expansion) were not so prominent, compared to those in disease activity indices, although there was statistical significance. Furthermore, the 3-month improvements in metrological indices (BASMI10 and chest expansion) were not correlated with those in BASDAI or ASDAS-CRP (data were not shown). These results may be related with the lack of association between the 3-month improvements in functional and metrological indices and long-term clinical remission achievement.
We have updated our manuscript with additional description in the “Discussion” section as follows; “The 3-month improvements in functional (BASFI) and metrological (BASMI10 and chest expansion) indices were not associated with long-term clinical remission achievement, which may be related with irreversible structural changes due to longer disease duration.”
- Commented [A15]: The CI is broad and includes 1.
[Response] Normalized CRP was only marginally associated with ASDAS-ID achievement. The odds ratio was 1.94 and confidence interval was 0.90-4.15 (P = .089). Therefore, we described as follows; “Normalized CRP, not ESR, at 3 months showed a marginally statistical significance (ESR, OR = 1.66; CI, 0.73–3.76; P = .226 and CRP, OR = 1.94; CI, 0.90–4.15; P = .089).”
- Commented [A16]: Please improve the sentence construction.
[Response] Thank you for your comments. We have revised the sentence as follows; “As younger age was associated with a higher achievement rate of both clinical remission criteria, we analyzed the proportion of attainment of ASDAS-ID and BASDAI-CRP using the age of 40 years as cut-off;”
- Commented [A17]: As improvement in 3 month BASDAI scores showed association with achievement of treatment target at 33 months, subgroup analysis was performed after cateogorisation according to BASDAI scores.
[Response] Thank you for your comments. We have revised the sentence as follows; “as improvement in 3 month BASDAI scores showed association with achievement of treatment target at 33 months, subgroup analysis was performed after cateogorisation according to BASDAI scores.”
- Commented [A20]: suggest improving the sentence construction.
[Response] Thank you for your comments. We have revised the sentence as follows; “With respect to age, patients aged <40 years showed two-times higher rate of achievement of both clinical remission criteria.”
- Commented [A22]: improve the sentence
[Response] Thank you for the suggestion. T2T strategy is now a well-established paradigm in the treatment of RA and is based on evidence developed in patients with early and established RA [10]. Clinical trials have provided consistent evidence that T2T strategy resulted in superior clinical outcomes among patients who are treated systematically with regular quantitative measurement of disease activity using continuous disease activity measures and enhanced long-term functional and quality of life outcomes. In addition, aiming for early remission in patients with RA is beneficial in the long-term in terms of better clinical and functional outcomes and lower healthcare costs [11]. In conclusion, rapid diagnosis and a T2T approach with tight monitoring and control, can increase the likelihood of remission in patients with RA [12].
We have updated with additional description as follows; “This treatment approach in RA is now a well-established practice with regular disease activity monitoring using validated composite measures of disease activity and remission criteria.”
- Commented [A23]: feasibility and cost effectiveness of T2T strategy is a concern. I am not sure about the word elucidation here. Please consider revising this sentence.
[Response] We appreciate the reviewer’s critical comments. In the T2T strategy, the clinician treats the patients aggressively enough to reach and maintain explicitly specified and sequentially measured goals, such as remission/inactive disease or low disease activity [13]. The biologics, which are by far the most effective currently available treatments across the axSpA spectrum, are still associated with very high treatment costs especially compared to NSAIDs and conventional synthetic disease modifying anti-rheumatic drugs (DMARDs) [14]. In RA, cost effectiveness as well as clinical effectiveness of T2T strategy has been evaluated [11, 15]. In T2T strategy of AS, the assessment in healthcare costs as well as clinical and functional outcome should be performed, which is what the authors were trying to explain about. However, feasibility and cost effectiveness of T2T strategy in AS was not described in this paper and the authors agreed to the reviewer’s comments. We have deleted the sentence following the reviewer’s comments.
- Commented [A27]: suggested (by who) would it be suggested by the task force or the cited studies.
[Response] We have added the words “by the task force”.
- Commented [A28]: as it comprises the estimation of clinical aspects of the disease as well as APRs.
[Response] Thank you for the suggestion. We have changed that sentence as follows; “as it comprises the estimation of clinical aspects of the disease as well as APRs.”
- Commented [A29]: targets or criteria?/ 16. Commented [A31]: targets of criteria?
[Response] Thank you for the comment. According to T2T strategy in AS, the treatment target should be clinical remission/inactive disease of musculoskeletal involvement, taking extra-articular manifestations into consideration. Clinical remission/inactive disease is defined as the absence of clinical and laboratory evidence of significant inflammatory disease activity. So, the authors thought that ‘criteria’ was a more appropriate word than ‘target’.
- Commented [A35]: we adopted the most strict cur off value-BASDAI<2 which was used in cohort and randomised controlled studies. Break the sentence.
[Response] Thank you for the comment. We have updated our manuscript in the “Discussion” section as follows; In this study, we adopted the most strict cur off value-BASDAI <2 which was used in cohort cohort and randomnised controlled studies. We found patients who satisfied the ASDAS-ID and BASDAI-CRP almost overlapped,”
- Commented [A38]: The variables discussed in this paper are not modifiable. Data on association between physical activity and remission might give an opportunity to patients in improving outcome. But that is not studied in this paper. The variables studied here might helping in choosing the patients who might benefit from changing therapy to attain treatment target.
[Response] We appreciate the reviewer’s critical comments. We have deleted that sentence in the “Discussion” section.
- Commented [A42]: Please restructure this sentence
[Response] Thank you for your comments. We have deleted the word “that” and updated in the “Discussion” section as follows; “As the clinical improvements at 3 months were the most prominent with TNFi treatment in our study, we tried to demonstrate early clinical improvements were associated with a higher possibility of sustained attainment of long-term clinical remission.”
- Commented [A43]: BASDAI also has several other components.
[Response] Thank you for your comments. In this sentence, we tried to focus on the impact of CRP normalization on achievement of both clinical remission criteria. ASDAS has its formula for calculation (ASDAS-CRP formula is 0.12 x Back Pain + 0.06 x Duration of Morning Stiffness + 0.11 x Patient Global + 0.07 x Peripheral Pain/Swelling + 0.58 x Ln(CRP+1)). ASDAS-ID achievement demands not only CRP improvement but also improvements in other four variables, while ASDAS-CRP achievement needs both CRP normalization and low BASDAI.
We have updated our manuscript in “Discussion” section as follows; “Normalized CRP at 3 months was an independent predictor of BASDAI-CRP, not ASDAS-ID, achievement, which may be related to the fact that BASDAI-CRP achievement needs both CRP normalization and low BASDAI, while ASDAS-ID achievement demands not only CRP improvement but also improvements in other four variables.”
- Comments for English Language Editing.
- Commented [A6]: sugeest replacing with "individual clinical parameters" - Commented [A7]: study period - Commented [A8]: 13 on adalimumab, suggest replacing "for" with "on" - Commented [A9]: males - Commented [A10]: years - Commented [A19]: In - Commented [A21]: rheumatic - Commented [A24]: widely - Commented [A25]: for - Commented [A26]: in AS - Commented [A30]: both ASDAS ID and BASDAI CRP - Commented [A32]: incorporates additional - Commented [A33]: clinical practice - Commented [A34]: strike to use - Commented [A36]: We found - Commented [A39]: beneficial - Commented [A40]: effect of baseline characteristics - Commented [A41]: criteria - Commented [A44]: that of |
[Response] We appreciate the reviewer’s comments. We have updated our manuscript following the reviewer’s comments.
References
- Zhang, S.; Li, Y.; Deng, X.; Huang, F. Similarities and differences between spondyloarthritis in Asia and other parts of the world. Curr Opin Rheumatol 2011, 23(4), 334-338.
- Rudwaleit, M.; Haibel, H.; Baraliakos, X.; et al. The early disease stage in axial spondylarthritis: results from the German Spondyloarthritis Inception Cohort. Arthritis Rheum 2009, 60(3), 717-727.
- Reveille JD: HLA-B27 and the seronegative spondyloarthropathies. Am J Med Sci 1998, 316, 239–249.
- Feltkamp, T.E.; Mardjuadi, A.; Huang, F.; Chou, C.T. Spondyloarthropathies in eastern Asia. Curr Opin Rheumatol 2001, 13(4), 285-290.
- Kim, T.J.; Kim, T.H. Clinical spectrum of ankylosing spondylitis in Korea. Joint Bone Spine 2010, 77(3), 235-240.
- Brophy, S.; Mackay, K.; Al-Saidi, A.; Taylor, G.; Calin, A. The natural history of ankylosing spondylitis as defined by radiological progression. J Rheumatol 2002, 29(6), 1236-1243.
- Ramiro, S.; van der Heijde, D.; van Tubergen, A.; et al. Higher disease activity leads to more structural damage in the spine in ankylosing spondylitis: 12-year longitudinal data from the OASIS cohort. Ann Rheum Dis 2014, 73(8), 1455-1461.
- Calvo-Gutierrez, J.; Garrido-Castro, J.L.; Gil-Cabezas, J.; et al. Is spinal mobility in patients with spondylitis determined by age, structural damage, and inflammation? Arthritis Care Res (Hoboken) 2015, 67(1), 74-79.
- Zochling J. Measures of symptoms and disease status in ankylosing spondylitis: Ankylosing Spondylitis Disease Activity Score (ASDAS), Ankylosing Spondylitis Quality of Life Scale (ASQoL), Bath Ankylosing Spondylitis Disease Activity Index (BASDAI), Bath Ankylosing Spondylitis Functional Index (BASFI), Bath Ankylosing Spondylitis Global Score (BAS-G), Bath Ankylosing Spondylitis Metrology Index (BASMI), Dougados Functional Index (DFI), and Health Assessment Questionnaire for the Spondylarthropathies (HAQ-S). Arthritis Care Res (Hoboken) 2011, 63 Suppl 11, S47-58.
- Smolen, J.S. Treat-to-target as an approach in inflammatory arthritis. Curr Opin Rheumatol 2016, 28, 297-302.
- Ten Klooster, P.M.; Oude Voshaar, M.A.H.; Fakhouri, W.; de la Torre, I.; Nicolay, C.; van de Laar, M.A.F.J. Long-term clinical, functional, and cost outcomes for early rheumatoid arthritis patients who did or did not achieve early remission in a real-world treat-to-target strategy. Clin Rheumatol 2019,38(10), 2727-2736.
- Burmester, G.R.; Pope, J.E. Novel treatment strategies in rheumatoid arthritis. Lancet 2017, 389(10086), 2338-2348.
- Machado, P.M.; Deodhar, A. Treat-to-target in axial spondyloarthritis: gold standard or fools' gold? Curr Opin Rheumatol 2019, 31, 344-348.
- Torgutalp, M.; Poddubnyy, D. Emerging treatment options for spondyloarthritis. Best Pract Res Clin Rheumatol 2018, 32(3), 472-484.
- Wailoo, A.; Hock, E.S.; Stevenson, M; et al. The clinical effectiveness and cost-effectiveness of treat-to-target strategies in rheumatoid arthritis: a systematic review and cost-effectiveness analysis. Health Technol Assess 2017, 21(71),1-258.